# Decoding the Impact of Tumor Microenvironment in Osteosarcoma Progression and Metastasis

**DOI:** 10.3390/cancers15205108

**Published:** 2023-10-23

**Authors:** Bikesh K. Nirala, Taku Yamamichi, D. Isabel Petrescu, Tasnuva N. Shafin, Jason T. Yustein

**Affiliations:** Aflac Cancer and Blood Disorders Center, Emory University, Atlanta, GA 30322, USA; bikesh.kumar.nirala@emory.edu (B.K.N.); taku.yamamichi@emory.edu (T.Y.); isabel.petrescu@emory.edu (D.I.P.); tasnuva.nuhat.shafin@emory.edu (T.N.S.)

**Keywords:** osteosarcoma, tumor microenvironment, therapeutic strategies, immune cells, nonimmune cells

## Abstract

**Simple Summary:**

Osteosarcoma (OS) is the most common bone tumor in the pediatric population. Currently, no effective molecularly targeted therapies are available for OS. The five-year survival rate of OS has increased to about 70% since the 1970s but is only 20–30% for patients with metastasis. The management of OS is challenging and requires a multidisciplinary approach. Surgical excision and systematic multiagent therapy are standard clinical practices for OS treatment. However, there is a pressing need to identify novel therapeutic approaches and biomarkers to manage the disease better. Understanding osteosarcoma’s tumor microenvironment (TME) has recently gained much interest towards providing valuable insights into tumor heterogeneity, progression, metastasis, and the identification of novel therapeutic avenues. In this review, we discuss the current understanding of the OS TME, including different cellular and noncellular components, their crosstalk with OS tumor cells, and their involvement in tumor progression and metastasis. Discovering more specific therapeutic targets, determining interactions among cellular and noncellular components of the OS TME, and using rational combination therapies targeting tumor intrinsic and TME features will inevitably improve OS patient outcomes.

**Abstract:**

Osteosarcoma (OS) is a heterogeneous, highly metastatic bone malignancy in children and adolescents. Despite advancements in multimodal treatment strategies, the prognosis for patients with metastatic or recurrent disease has not improved significantly in the last four decades. OS is a highly heterogeneous tumor; its genetic background and the mechanism of oncogenesis are not well defined. Unfortunately, no effective molecular targeted therapy is currently available for this disease. Understanding osteosarcoma’s tumor microenvironment (TME) has recently gained much interest among scientists hoping to provide valuable insights into tumor heterogeneity, progression, metastasis, and the identification of novel therapeutic avenues. Here, we review the current understanding of the TME of OS, including different cellular and noncellular components, their crosstalk with OS tumor cells, and their involvement in tumor progression and metastasis. We also highlight past/current clinical trials targeting the TME of OS for effective therapies and potential future therapeutic strategies with negligible adverse effects.

## 1. Introduction

Osteosarcoma (OS) is a heterogeneous, highly metastatic bone malignancy with complex genetic and chromosomal alterations, most frequently occurring in children and young adults [1,2,3]. The genetic landscape of OS is characterized by a high degree of chromosomal instability that leads to aneuploidy and recurrent somatic mutations. The copy number loss at chromosomes 3q, 6q, 9, 10, 13, 17p, and 18q and the copy number gain at chromosomes 1p, 1q, 6p, 8q, and 17p have been reported in OS tumors [4]. Previous studies have shown that the inactivation of tumor suppressor genes, such as *TP53*, *RB1*, and *ATRX*, and the amplification of oncogenes, including *MYC* and *MDM2*, are associated with OS pathogenesis [4,5,6,7,8]. These genetic alterations can influence the tumor microenvironment (TME) and contribute to the aggressive behavior of OS.

Despite progress with multimodal treatment approaches, patient outcomes remain poor for recurrent and metastatic patients, with a five-year survival rate of less than 30%. The lung is the most common site of metastasis, followed by the bones [3]. Unfortunately, no durable, effective molecularly targeted therapy for the disease is currently available. Understanding OS’s TME has recently gained much attention among scientists, hoping to provide valuable insights into tumor heterogeneity, progression, and metastasis to target this aggressive tumor type. OS tumors grow in a complex and dynamic bone microenvironment consisting of bone cells, stromal cells, vascular cells, immune cells, and mineralized extracellular matrix (ECM). The complex interplay between OS cells and their adjoining microenvironment plays a critical role in tumor progression, apoptosis, invasion, metastasis, angiogenesis, creation of the pre-metastatic niche, and response to therapy [9,10]. 

The immune landscape of OS is particularly intriguing. Emerging evidence suggests that the immune system can promote or inhibit OS progression, depending on the context and cell type [10,11,12]. For instance, specific immune cells, such as T cells, B cells, and natural killer (NK) cells, can exert antitumor effects, while others, like myeloid-derived suppressor cells (MDSCs), M2 macrophages, and regulatory T cells (Tregs), can promote tumor growth and metastasis. Thus, understanding the intricate interactions between OS and immune cells is crucial for developing effective immunotherapies [10,13].

In addition to the cellular components, noncellular elements of the TME, such as the ECM and extracellular vesicles (EVs), also contribute to OS progression and metastasis. The ECM, a three-dimensional network of proteins and polysaccharides, provides a scaffold for tumor cells, influences their behavior through biomechanical and biochemical cues, and is crucial for modeling and studying drug response [14,15]. By comparison, EVs, tiny membrane-bound particles released by cells into the extracellular matrix, serve as vehicles for intercellular communication, carrying a cargo of proteins, lipids, and nucleic acids that can alter the phenotype of recipient cells. Additionally, EVs are involved in cell communication, migration, angiogenesis, and tumor growth [16]. EVs have been shown to play an essential role in OS development, progression, and metastatic processes [17,18,19]. The complexity and diversity of the OS TME exhibit both challenges and prospects for therapeutic intervention. Emerging strategies aim to target the TME in isolation or combined with conventional therapies in order to disrupt the supportive network that tumor cells rely on for survival and spread. These include immunotherapies designed to reactivate antitumor immune responses, antiangiogenic agents to normalize the aberrant vasculature, and drugs that target the ECM or EVs to disrupt tumor–stroma crosstalk.

In this review, we aim to provide a comprehensive overview of the current understanding of the OS TME, focusing on its role in tumor progression and metastasis. We will delve into the various cellular and noncellular components of the TME, discuss their roles in OS, and explore potential therapeutic strategies that target the TME. By shedding light on this complex landscape, we hope to pave the way to develop more effective treatment strategies for OS.

## 2. The Tumor Microenvironment of OS

The OS tumor grows in a very complex and dynamic bone microenvironment consisting of cellular and noncellular components, including bone cells (osteoblasts, osteoclasts, and osteocytes), stromal cells (mesenchymal stem cells and fibroblasts), vascular cells (endothelial cells and pericytes), immune cells (myeloid and lymphoid cells), and mineralized extracellular matrix (ECM), as shown in Figure 1. Myeloid cells are the most abundant cell type in the TME of OS. A single-cell analysis showed multiple ligand–receptor interactions between OS tumor, myeloid, and osteoblast cells, including 21 ligand–receptor gene pairs that are significantly associated with survival outcomes [20]. The TME not only provides favorable conditions for tumor cell growth but also releases a range of elements, including various cytokines, chemokines, and growth factors, that can promote the metastasis of tumor cells to other tissues and organs [21]. Figure 2 shows the interaction between tumor cells and various immune cells, including the myeloid and lymphoid cells in the TME of OS.

### 2.1. Bone Cells

Three different bone cells contribute to OS’s bone homeostasis: osteoblast cells, osteoclast cells, and osteocytes. Osteoblast cells are bone-forming, whereas osteoclast cells are bone-resorbing, and osteocytes are mature bone cells.

#### 2.1.1. Osteoblast Cells

Osteoblasts (OBs) are bone-forming cells originating from the pluripotent mesenchymal stem cells. Specific cytokines promote osteoblastogenesis, including interferon-γ (IFN-γ), interleukin-10 (IL-10), IL-11, IL-18, cardiotrophin-1, and oncostatin M. In contrast, IL-4, IL-7, IL-12, IL-13, IL-23, tumor necrosis factor-α (TNF-α), TNF-β, IL-1α, IFN-α, and IFN-β obstruct it [22]. The expression of Runt-related transcription factor 2 (Runx2) is essential for OB differentiation and is upregulated in the committed mesenchymal stem cells and the pro-osteoblasts. *RUNX2* regulates *FGFR2* and *FGFR3,* which are necessary for the proliferation of osteoblast progenitors. In addition, *RUNX2* governs Hedgehog, Wnt, and Pthlh (*Pthr1*) signaling pathway genes, which induce pro-osteoblasts to commit to the osteoblast lineage [23,24]. Recently, it has been found that Krüppel-like factor 2 (*KLF2*), a zinc-finger DNA-binding transcription factor, regulates osteogenic differentiation and bone mineralization mediated through *RUNX2* [25,26]. 

OBs can affect the formation, differentiation, or apoptosis of osteoclasts (OCs) through several pathways, including the OPG/RANKL/RANK, RANKL/LGR4/RANK, Ephrin2/ephB4, and Fas/FasL pathways [27]. OBs communicate with OS through OB-derived extracellular vesicles (OB-EVs) to regulate the TME of OS [28]. 

#### 2.1.2. Osteoclast Cells

The OCs are specialized cells responsible for bone resorption that originate from myeloid precursor cells [20]. OCs communicate closely with osteoblasts and osteocytes under normal physiological conditions and play a vital role in maintaining bone homeostasis [29,30]. Under pathological conditions, OCs play an active role in OS tumorigenesis and metastasis [31]. The cytokines colony-stimulating factor 1 (CSF1) and receptor activator of nuclear factor kappa B ligand (RANKL) are essential and sufficient for the differentiation and activation of OCs [32,33,34]. The cytokine CSF1 is involved in the proliferation and survival of pre-osteoclast (POC) cells. RANKL drives the differentiation of OC precursor cells into mature OCs by controlling gene expression through the activation of its receptor, RANK [35]. OS cells regulate OC migration, resorption, and osteoclastogenesis by secreting EVs, which is mediated through the regulation of the PTEN/PI3K/AKT pathway, miR-19a-3p, and the cytokines CSF1 and RANKL [21,36].

The role of OCs in OS tumor progression and metastasis is still an active area of investigation. Immunohistochemistry (IHC) staining and mRNA expression of TRACP5, an OC cell surface marker, were relatively low in OS patient samples compared to healthy controls, which suggests a negative correlation between OS development and a high OC population. Additionally, higher OC-retained patients exhibit better chemotherapy efficacy compared to patients with a low population of OCs in the TME of OS [37]. However, the underlying mechanism of the inhibition of generation and maturation of OCs by chemoresistant malignant tumors remains to be explored. Compared to primary osteoblastic OS lesions, lower OC infiltration is observed in chondroblastic, recurrent, and lung metastatic samples of OS lesions [38]. Li et al. found that OCs can function like antigen-presenting cells (APCs), similar to dendritic cells, to activate CD4^+^ and CD8^+^ T cells [39].

Interestingly, OCs can exhibit antitumor activity, while several reports have also demonstrated their tumor-promoting functions. Li et al. revealed that cell–cell communication between OCs and CD4^+^ Tregs alters the TME of OS, which is associated with a poor prognosis [40]. In one study, zoledronic acid (ZA) treatment enhanced the antitumor effect of cisplatin on OS by targeting the ROS-PI3K/AKT signaling pathways, which suggests a possibility for combining ZA with other chemotherapeutic reagents, such as cisplatin, as a target against OS [41]. However, contrary results were reported in several clinical studies where the loss of OCs using ZA showed more highly metastatic disease than in the control group. In contrast, enhancing the OC population using fulvestrant treatment reduces metastatic lesions [42,43]. It is hypothesized that the role of OCs varies depending on the stage of OS progression. In the primary tumor, OCs may provide a niche within the bone that nurtures the OS cells and suppresses metastasis. In the later stages, loss of OCs may favor metastatic spread [44]. More extensive studies are needed to understand the pathophysiological role of OCs in OS progression and metastasis before using OCs as a therapeutic target for OS patients.

#### 2.1.3. Osteocytes

Osteocytes are mature, mineralized bone cells that are derived from osteoblast cells. They are the most abundant cell type in bone, with an estimated half-life of 25 years. Previously, osteocytes were believed to be passive cells. However, multifunctional roles have recently been reported, including coordinating bone formation and reabsorption, sensing the mechanical force in bone, etc. Osteocytes communicate with OS cells by secreting several soluble factors, including GDF15, TGFβ, CXCL1/2, and VEGFA, as well as through physical interactions (e.g., NOTCH3 signaling). Osteocytes also produce RANKL, CSF1, HMGB1, and IL-11 to stimulate osteoclastogenesis and bone resorption [45]. Osteocytes activate the CXCL12-CXCR4 signaling axis in OS by producing CXCL12, favoring tumor metastasis. Although osteocytes can potentially establish bone metastasis and homing to specific areas of the bone, further research is required to determine osteocytes’ particular contribution to the relocation and homing of cancer cells to bone. 

### 2.2. Mesenchymal Stem Cells

Mesenchymal stem cells (MSCs) and osteoblast cells are considered potential precursors of OS cells [46]. Studies have shown that MSCs act as sensors between OS tumor cells and the TME by releasing several cytokines, chemokines, interleukins, and EVs. MSCs actively participate in paracrine signaling with OS tumor cells, impacting diverse aspects of tumor behavior, including angiogenesis, proliferation, invasion, metastasis, immune modulation, and chemotherapeutic resistance [46]. MSCs promote OS growth, metastasis, and angiogenesis by secreting various chemokines, including chemokine ligand 5 (CCL5), stromal-derived factor 1 (SDF-1), CXCL12, IL-6, and VEGF [46]. The EVs released by both OS cells and MSCs mediate intercellular communication by transporting miRNAs/RNAs and proteins. Bone marrow mesenchymal stem cell-derived extracellular vesicles (BMSC-EVs) promote OS cell proliferation, invasion, and migration via the MALAT1/miR-143/NRSN2/Wnt/β-catenin axis and miR-655-mediated β-catenin signaling [47,48]. A cytokine-like hormone, leptin, highly expressed in MSCs, shapes the OS TME by enhancing autophagy, promoting chemoresistance and OS cell survival, mediated through TGF-β upregulation [49]. Additionally, BMSC-derived exosomes have been implicated in promoting OS tumorigenesis and metastasis by inducing autophagy [50]. Although MSCs have the potential for therapeutic interventions, such as bone regeneration and engineered protein delivery, there is continued debate about their precise role in OS development, owing to complexities in differentiating MSCs from osteoblasts and understanding their dual role [51]. Utilizing the potential of MSCs, engineered extracellular microvesicles encapsulating chemotherapeutic drugs, including doxorubicin, have emerged as a promising nanocarrier for targeted OS therapy, exhibiting superior specificity, biocompatibility, and reduced toxicity [52]. Furthermore, exosomal microRNAs from bone marrow-derived MSCs contribute to a dual role in OS, with miR-208a and miR-21-5p promoting proliferation, migration, and invasion [53,54]. Concurrently, miR-206 inhibits OS progression by targeting TRA2B, highlighting potential therapeutic avenues [55].

### 2.3. The Immune Landscape of OS

#### 2.3.1. Lymphoid Cells

Lymphoid cells originate from hemopoietic stem cells. There are three major types of lymphocytes: B lymphocytes or B cells, T lymphocytes or T cells, and natural killer cells (NK cells). B cells develop in the bone marrow and are responsible for antibody production. T cells mature in the thymus and have cell surface receptors to recognize antigens bound to the major histocompatibility complex (MHC). NK cells arise from bone marrow; are stimulated by specific cytokines, including interferon-gamma; and recognize and attack “non-self” cells, including cancer cells. Lymphoid cells play an essential role in OS tumorigenesis and metastasis. Here, we discuss the role of different lymphoid cell types in OS tumor progression and metastasis.

##### T Cells

T cells are essential for both cellular and humoral immunity. Their infiltration of the TME plays a critical role in antitumor immunity in several tumor types, including OS [10,11,56]. Heterogeneous T cell subsets in the TME of OS have distinct functions: cytotoxic T cells (CD8^+^) directly attack cancer cells, whereas helper T cells (CD4^+^) orchestrate the immune response, and regulatory T cells (Tregs) suppress the immune response, promoting tumor growth [57]. Recent studies have shown a complex interplay between tumor cells and T cells in the TME of OS. Wu et al. conducted a comprehensive analysis, revealing a complex immune landscape in OS with diverse immune cell infiltration and checkpoint expression [58]. Liu et al. discovered an immune evasion mechanism where the OS cells express PD-L1 and block T cell function, evading immune surveillance [11].

The TME of OS not only influences T cell function and immunosuppression but can also impact T cell trafficking and infiltration. Factors such as chemokines, adhesion molecules, and the extracellular matrix can regulate T cell migration into the TME [10,56]. The chemokine CXCL12 and its receptor CXCR4 have been implicated in T cell trafficking in OS [10]. Therefore, strategies to enhance T-cell infiltration could potentially improve the efficacy of T cell-based therapies [59]. However, the effectiveness of T cell-based therapies in OS is often limited by the immunosuppressive TME, especially by the infiltration of Tregs and myeloid-derived suppressor cells (MDSCs) [56,58]. Therefore, strategies to modulate the TME and enhance T cell function are crucial for improving the efficacy of T cell-based therapies in OS.

##### B Cells

B cells are vital to adaptive immunity and significantly impact the OS TME by producing antibodies, presenting antigens, and secreting cytokines. Most of the literature suggests that B cells promote tumor progression; however, recent investigations have demonstrated the close association of B cells with the prognosis of OS patients and the response to immunotherapy [15,18]. Recently, Petitprez et al. reported that B cells were the most decisive prognostic factor in improved survival and a high response rate to PD1 blockade in a phase 2 clinical trial of soft-tissue sarcoma [60,61]. On the contrary, Kendal et al. explored the dual role of B cells in cancer progression, highlighting their capacity to either promote tumor growth by secreting immunosuppressive cytokines and promoting regulatory T cell formation or to inhibit tumor growth through antigen presentation and antibody production [62]. Moreover, the function of B cells in OS is closely related to the presence and function of other immune cells. Circulating follicular helper T cell (Tfh) abnormality leads to altered B cell maturation and differentiation that is displayed in OS patients. The Tfh cells are crucial for B cell maturation and antibody production [63]. This study suggested that the interaction between Tfh cells and B cells could be a potential therapeutic target in OS. Wang et al. investigated the effect of miR-138 on the function of Tfh cells and the differentiation of B cells in OS [64]. This study found that miR-138 could regulate the function of Tfh cells, thereby affecting the differentiation of B cells. This finding further underscores the intricate relationship between B cells and other immune cells in the context of OS [65]. Furthermore, Bod et al. discussed the role of B cell-specific checkpoint molecules in regulating antitumor immunity [66]. They found that certain checkpoint molecules expressed on B cells can inhibit B cell function and promote tumor growth. Therefore, targeting these checkpoint molecules could enhance the antitumor activity of B cells. Recent research underscores B cells’ complex role in OS. Zhang et al. used single-cell RNA sequencing to investigate the effect of B cells on the prognosis and immune cell infiltration of OS. Two of the B cell marker genes in OS, *RPL37A* and *MEF2C*, were associated with a poor prognosis, while the marker genes *PLD3* and *SNX2* were associated with a good prognosis [67].

B cells influence disease progression, interact with other immune cells, and hold promise as prognostic markers. Further research is needed to understand the significance of their role in OS progression and the development of novel therapeutic strategies.

##### NK Cells

Natural killer (NK) cells, critical components of the innate immune system, play a vital role in the immune response against OS by recognizing and eliminating malignant cells without prior sensitization [68,69,70]. NK cells are characterized by their cytotoxic activity and the production of cytokines, such as interferon-gamma (IFN-γ), which can modulate the TME and enhance antitumor responses. However, the TME of OS is complex and can influence the function and efficacy of NK cells [71,72]. A study by Liu et al. revealed a novel immune evasion mechanism in OS at the single-cell level, noting that the TME of OS is characterized by a high degree of heterogeneity, with NK cells comprising a considerable proportion of immune cells in the TME. Despite their high presence, NK cells in the TME of OS often exhibit functional exhaustion, characterized by decreased cytotoxic activity and cytokine production [11]. This exhaustion can be attributed to various factors, such as dysregulated NK cell receptor signaling, the immunosuppressive effect of regulatory cells in the TME, the interaction with other immune cells, the expression of immune checkpoint molecules, and soluble factors in the microenvironment [11,56,57,73].

Recent studies have highlighted the potential of NK cell-based immunotherapies in OS. For instance, Rademacher et al. demonstrated that overexpression of interleukin-12 (IL-12) in sarcoma cells can enhance NK cell immunomodulation, leading to increased cytotoxicity against tumor cells [74]. Similarly, Omer et al. emphasized the potential of NK cell targeting in pediatric sarcoma, suggesting that enhancing NK cell function could improve therapeutic outcomes [75]. Moreover, Chu et al. showed that the combination of N-803 (an IL-15 superagonist) and dinutuximab (an anti-GD2 antibody) with ex vivo expanded NK cells significantly enhances in vitro cytotoxicity against GD2^+^ pediatric solid tumors and improves the survival of xenografted immunodeficient NSG mice [76]. This suggests that strategies aimed at improving the function and survival of NK cells could be beneficial in OS treatment. Another promising approach is the inhibition of TIGIT, an immune checkpoint molecule expressed in NK cells. Judge et al. showed that the combination of IL-15 stimulation and TIGIT blockade significantly enhances the antitumor activity of NK cells in soft-tissue sarcomas [77]. Furthermore, a study by Lu et al. revealed that panobinostat, a histone deacetylase inhibitor, can enhance NK cell cytotoxicity in soft-tissue sarcomas, suggesting its potential use in combination with NK cell-based immunotherapies [78].

Although NK cells hold great promise for future OS treatment, several challenges, including the heterogeneity of the OS TME, the immunosuppressive mechanisms employed by the tumor, and the potential toxicity of NK cell-based therapies, need to be carefully considered when designing future therapeutic strategies. 

#### 2.3.2. Myeloid Cells

Myeloid cells derive from hematopoietic stem cells in the bone marrow [79], which include granulocytes, monocytes, macrophages, and dendritic cells (DCs) [80]. Myeloid cells in OS have various roles for tumor development and metastasis, depending on the cell type. Such roles include phagocytosis, inflammation, migration, or the adaptive immune system’s activation of a T cell response [81]. 

##### Monocytes

Monocytes are antigen-presenting cells (APCs) that can communicate between the innate and adaptive immune systems. This cell type can differentiate into macrophages or dendritic cells [81]. Monocytes are one of the cell types that produce the chemokine monocyte chemoattractant protein-1 (MCP-1), which is associated with tumor cell growth, migration, invasion, and metastasis in several types of cancer. In vitro studies revealed a strong correlation between MCP-1 production and cell migration in OS [82]. Monocytes expressing the CCR2 receptor are recruited to sites of metastases by the chemoattractant cytokine CCL2, suggesting that this interaction can provide a mechanism for the targeted treatment of pulmonary metastasis. Losartan is an angiotensin II type I receptor (AT1R) antagonist found to inhibit CCR2 signaling, resulting in reduced monocyte recruitment [83]. In canine models of lung metastatic OS, combined treatment with high-dose losartan and the tumor-targeted multikinase inhibitor toceranib effectively reduced advanced lung metastases in 50% of the dogs receiving treatment [84]. Furthermore, patrolling monocytes (PMOs) are a monocyte subtype exhibiting antitumor activity in OS. PMOs are enriched in the pulmonary microvasculature and have been shown to prevent cancer cell migration and promote the recruitment and activation of natural killer cells [85].

##### Macrophages

Macrophages are some of the most abundant immune cells in the TME of OS and play multifunctional roles in tissue homeostasis, host defense, tissue repair, and apoptosis by releasing various growth factors, cytokines, chemokines, and enzymes [86,87]. The role of macrophages in OS tumor progression shows contrasting results depending on their polarization. These highly plastic cells can be influenced by external stimuli and their surrounding environment and polarize into M1- or M2-like macrophages. M1-like macrophages participate in anticancer adaptive immunity, whereas M2-like macrophages promote the immune-suppressive TME. Recently, we noted that hyperactivation of Myc reduces the macrophage population in the TME of OS in murine models [6,88,89]. The human TARGET dataset also showed a reduction in the enrichment of the macrophage population when MYC is highly expressed (https://ocg.cancer.gov/programs/target (accessed on 25 September 2023)). A high macrophage population was also associated with a good prognosis for OS patients [90]. Macrophages play an essential role in inhibiting the initiation and development of OS. Reorienting and polarizing tumor-associated macrophages toward M1-like macrophages is the holy grail of macrophage-mediated cancer therapy [91,92]. Due to their plasticity and heterogeneity, tumor-associated macrophages (TAMs) have shown both pro- and antitumor activity according to the tumor type and the interactions between TAMs and other cell types in the TME [93,94]. Emerging studies have found that TAMs can mediate immunosuppression via interaction with various immune effector cells. It is reported that TAMs express the ligand PD-1 and cytotoxic T lymphocyte-associated antigen-4 (CTLA-4), inhibiting T cell activation. Studies have found that TAMs can also be involved in the recruitment of Treg cells into tumor tissues and in establishing pre-metastatic niches to the distal organs [95,96]. TAMs are also involved in the secretion of various pro-angiogenic factors, such as vascular endothelial growth factor (VEGF), fibroblast growth factor (FGF), and matrix metallopeptidase 9 (MMP-9), associated with tumor angiogenesis [97]. Before using TAMs as a therapeutic target, further study on the roles of macrophages in OS progression and metastasis must be accomplished. Macrophages are grouped into two major categories: the subtype that releases proinflammatory cytokines that lead to antimicrobial or antitumor activity is called the M1-macrophage. The other macrophage subtype that releases anti-inflammatory cytokines and supports tumor growth, invasion, and metastasis is called the M2-macrophage [81].

M1-Macrophages

M1 macrophages produce inducible nitric oxide synthase (iNOS) and express cytokines that induce the development of type-1 helper T cells [98]. These proinflammatory cytokines include interleukin-1 beta (IL-1β) and tumor necrosis factor-alpha (TNF-α) [99]. Once activated, these macrophages secrete chemokines that target proliferating tumor cells [100]. M1 macrophage-related genes can be biomarkers to assess high- and low-risk groups for individuals with OS. These genes are *CD37*, *GABRD*, and *ARHGAP25*, with roles in tumorigenesis and disease progression [99]. 

M2-Macrophages

M2 macrophages are involved in immune suppression, tissue remodeling, tumor progression, and angiogenesis [98]. M2 macrophages facilitate OS metastasis by secreting CCL18, MMP-12, cyclooxygenase 2 (COX-2), and IL-1β [101,102,103]. T cell immunoglobulin and mucin domain (Tim) family of protein, Tim-3, was found to promote M2 macrophage polarization, leading to invasion and metastasis of OS cells [104]. Han et al. identified that IL-1β secreted by M2 macrophages contributes to OS metastasis, mediated through the NF-κB/miR-181α-5p/RASSF1A/Wnt pathway [105]. Aberrant methylation at the promoter of RASSF1A, or Ras association domain family 1A, contributes to the pathogenesis of several types of pediatric tumors, including OS. A potential therapeutic method to minimize M2 macrophage polarization is treatment with all-*trans* retinoic acid (ATRA). ATRA can directly inhibit M2 macrophage polarization to delay OS initiation by inhibiting colony and osteosphere formation [106]. Since the repolarization of M2-like macrophages to M1-like macrophages could be achieved with various strategies, this is emerging as an innovative anticancer approach [100]. When M2 macrophages are treated with the terpenoid compound asiaticoside (ATS), the expression of M2 macrophage markers is reduced, with no change in the expression of M1 markers, suggesting that treatment with ATS can repolarize M2 macrophages to M1 macrophages [107]. Alternatively, graphene oxide (GO), used for photothermal therapy (PTT), and mifamurtide, a synthetic analog of the cell wall in bacteria, have been shown to repolarize macrophages and suppress the progression of OS [100].

##### Dendritic Cells

Dendritic cells (DCs) act as professional antigen-presenting cells (APCs) that contribute to antigen-specific adaptive immunity with a role in T-cell priming, activation, and differentiation [108]. Depending on their origin, four distinct subclusters of DCs have been identified in OS lesions: monocyte-derived CD14^+^CD163^+^ DCs, the conventional myeloid-derived CD1^+^ DCs (cDC2s), CD141^+^CLEC9A^+^ DCs (cDC1s), and the activated CCR7^+^ DCs [38]. DCs produce interferon type 1α (IFN-1α) and IFN-1β, which are associated with antitumor and antiviral response [109]. Kawano et al. revealed that a murine OS model treated with doxorubicin and DCs exhibited increased levels of immunological cell death and expression of calreticulin (CRT), heat shock promoter 70 (HSP70), and high mobility group box 1 (HMGB1), which are involved in the systemic immune response. In addition, combined treatment resulted in increased cytotoxic T cells within metastases, inhibiting metastatic growth [110]. Single-cell RNA sequencing (scRNA-seq) data showed that a cluster of regulatory DCs might shape the immunosuppressive microenvironment in OS by recruiting regulatory T cells [11]. Similarly, murine models of metastatic OS that were treated with DCs exposed to cryo-treated tumor lysates and injected with anti-transforming growth factor-β (anti-TGF-β) antibody demonstrated reduced metastatic tumor volume, decreased regulatory T cells, and an increase in cytotoxic T cells [111]. As another potential therapeutic target, the transcription factor recombination signal binding protein for immunoglobulin kappa J region (RBP-J) is crucial for regulating DC synthesis and is involved in DC-dependent antitumor immune responses. Furthermore, when photodynamic therapy (PDT) is used as a treatment for OS, this improves the function of DCs for antigen processing and presentation to activate T cell-mediated immunity by upregulating HSP70 [112]. These studies provide meaningful insights into the regulatory pathways contributing to the antitumor response of DCs in OS. OS tumors treated with DCs combined with anti-glucocorticoid-induced tumor necrosis factor receptor (GITR) antibodies produce reduced levels of immunosuppressive cytokines and show antitumor effects. Moreover, a single-cell RNAseq study showed that a high infiltration of resting DCs in OS tumors was associated with poor prognosis [113].

##### Neutrophils

Tumor-associated neutrophils (TANs) are strongly correlated with tumor progression and metastasis in OS. TANs have been found to infiltrate primary OS as well as metastatic and recurrent OS [114]. Tokgöz et al. revealed that the neutrophil-to-lymphocyte ratio (NLR) in the peripheral blood of OS patients provides diagnostic and prognostic value, with a high NLR being associated with poor prognosis. Conversely, a low NLR indicates a better response to neoadjuvant chemotherapy and a higher pathological complete response rate [115]. Minimizing the presence of TANs in the TME of OS to reduce tumor progression and metastatic potential offers a promising therapeutic strategy.

##### MDSCs

Myeloid-derived suppressor cells (MDSCs) are immature myeloid cells that are pivotal in tumor-associated immune suppression by differentiating into tumor-associated DCs, TAMs, and TANs [116]. MDSCs can suppress the innate and acquired immune system by suppressing T cells, NK cells, and dendritic cell functions while activating FoxP3^+^ Treg cells [117]. The cytokine IL-17 could be involved in the immune-suppressive function of MDSCs on T cells through the upregulation of ARG-1, MMP-9, indoleamine 2,3-dioxygenase (IDO), and COX-2 [118]. MDSCs interact closely with T lymphocytes and prevent T cell-mediated immunity by producing reactive oxygen species (ROS), ablating L-arginine, and suppressing T cell proliferation [117]. MDSCs can also generate ROS by upregulating nicotinamide adenine dinucleotide phosphate (NADP) oxidase and increasing STAT-3 activation. In addition, MDSCs hinder the antigen presentation of DCs and induce phagocytosis of NK cells, leading to immune suppression [119]. Furthermore, MDSCs influence tumor progression and metastasis by forming a pre-metastatic niche, promoting angiogenesis and tumor cell invasion by producing higher levels of FGF, VEGF, VEGF analog Bv8, and MMP-9 [116,120,121]. A study of leukocyte populations, including myeloid cell types, in canine models of OS revealed that percentages of polymorphonuclear myeloid-derived suppressor cells (PMN-MDSCs) and monocytic (M-) MDSCs were increased in dogs with OS [122]. Shi et al. identified the compound (*S*)-(−)-N-[2-(3-Hydroxy-1H-indol-3-yl)-methyl]-acetamide (SNA) isolated from the plant *Selaginella pulvinata* as an inhibitor of phosphatidylinositide 3-kinases (PI3Ks), which are known to activate MDSCs. SNA treatment of in vivo murine models of OS demonstrated decreased functional activity of MDSCs. Combination treatment with SNA and anti-programmed cell death-1 (PD-1) antibody slowed OS tumor growth and improved survival by inhibiting MDSCs [123]. Thus, targeting MDSCs is an emerging therapeutic approach for OS.

### 2.4. Mast Cells

Mast cells in OS contribute to poor prognosis and tumor progression by promoting angiogenesis, extracellular matrix degradation, and tissue remodeling by producing angiogenic factors and proteases [124]. Lei et al. have identified a distinct nine-gene signature (*ZFP90*, *UHRF2*, *SELPLG*, *PLD3*, *PLCB4*, *IFNGR1*, *DLEU2*, *ATP6V1E1*, and *ANXA5*) that can accurately predict OS outcome and is strongly linked to activated mast cells [125]. Mast cells are primarily concentrated and sustained by cancer cells at the bone–tumor interface in OS. Mast cells are involved in local inflammation, immune cell recruitment, and osteoclastogenesis, indicating their role in OS progression and tissue repair. Mast cells potentially contribute to osteolysis through RANKL secretion [61]. In OS patients, activated mast cells display a complex phenomenon. Their presence is positively correlated with activated NK cells; however, mast cells are negatively correlated with M2 macrophages and memory B cells. Additionally, reduced infiltration of activated mast cells and DCs is associated with improved prognosis in OS patients [126]. Therefore, it is evident that the role of mast cells in shaping the OS microenvironment by regulating macrophage subtypes (M0, M1, and M2) and their impact on disease prognosis highlights their relevance as a potential therapeutic biomarker [126].

### 2.5. The Vasculature of the OS

Vascularization is driven by sprouting neoangiogenesis involving endothelial progenitor cells (EPCs), which play a vital role in OS growth and dissemination. Amplified VEGF pathway genes, particularly VEGF-A, are associated with advanced tumor stages and metastasis [127]. Hypoxic and acidic bone microenvironments contribute to the expression of angiogenic factors, while aggressive OS cells may utilize vascular mimicry, forming vasculogenic microchannels [128,129]. Tumor-derived EVs, through their cargo containing angiocrines and angiogenesis-related miRNAs, facilitate intercellular communication and promote angiogenesis [130]. Therapeutically targeting neovascularization in OS has been explored through clinical trials using antiangiogenic agents, especially to inhibit VEGFRs, yielding promising results in increasing progression-free survival [131]. However, challenges remain in effectively targeting tumor cells and the vascular microenvironment, as demonstrated by the need for comprehensive strategies such as VEGFR-2 inhibition combined with other therapies. Park et al. targeted the tumor vasculature using the VEGF blockade, which showed enhanced high endothelial venules in the TME and substantially enhanced T cell infiltration, significantly improving the therapeutic efficacy in preclinical models [132]. Novel biomimetic scaffolds designed for coupled angiogenesis and osteogenesis hold potential for tissue engineering applications, expanding possible options for treating bone malignancies and defects [133]. Exosomes, acting as regulators of angiogenesis, downregulate exosomal lncRNA *OIP5-AS1*, potentially affecting angiogenesis through ATG5- and mir-153-mediated autophagy, suggesting new avenues for understanding tumor development and prognosis [130]. The convergence of these insights highlights the multifaceted impact of vasculature on OS behavior, indicating innovative strategies for therapeutic intervention and prognosis prediction. 

### 2.6. The ECM of the OS Tumor

OS is characterized by its hallmark production of a rich ECM that influences tumor behavior and therapeutic responses [14,134,135]. The ECM includes components such as collagens, fibronectin, and laminins that contribute to aberrant signaling and structural abnormalities that promote sarcoma growth and also contribute to metastasis through distinct and interrelated mechanisms, including the interaction with cellular receptors associated with signaling pathways [14]. The ECM also modulates the insulin-like growth factor (IGF) axis, which regulates OS growth and resistance to conventional therapies [134]. Moreover, emerging research explores the potential for targeting the ECM in OS treatment. Hydrogels, resembling the ECM, can deliver therapeutic agents, offering a platform for OS therapy and bone regeneration [136]. Furthermore, the disruption of ECM-related factors such as neural EGF-like molecule 1 (NELL1) shows potential as a novel therapeutic approach to hinder OS progression [135]. Meanwhile, the crucial role of matrix metalloproteinases (MMPs) and their inhibition, exemplified by doxycycline treatment, reveals a promising chemotherapeutic avenue to curb OS aggressiveness by preventing ECM degradation and inhibiting angiogenesis [16].

The ECM interacts with the TME and cancer-associated fibroblasts (CAFs), contributing to OS development [135,137]. ECM components, such as EVs, are potential biomarkers for OS diagnosis and prognosis [138]. Novel approaches, such as 3D bioprinting with functionalized hydrogels, hold potential for targeted therapies by creating artificial microenvironments for improved treatment strategies [139]. Innovative approaches like attIL12-T cells also show promise by targeting CAFs within the ECM, disrupting tumor stroma, and favoring T-cell infiltration for enhanced therapy [137]. In conclusion, the significant role of the extracellular matrix in OS makes it a viable focus for therapeutic strategies. 

### 2.7. Extracellular Vesicles in OS (EVs)

Tumor cells produce EVs that contain cargo, such as microRNA, RNA, and proteins, to communicate with surrounding cells. This can contribute to an immunosuppressive TME [140]. A study by Luong et al. revealed that EVs secreted by OS cells and taken up by monocytes decreased the expression of proinflammatory cytokines and increased the expression of suppressive cytokines and other suppressive molecules. Research into circulating nucleic acid sequences associated with EV preparations identified a greater representation of repetitive element sequences in sera collected from OS patients at diagnosis vs. controls [141]. A study into the contribution of EVs for pre-metastatic niche (PMN) formation in OS demonstrated that tumor-derived EVs are insufficient to establish the PMN. Therefore, EVs were proposed to function alongside soluble tumor-secreted factors to regulate metastatic OS progression and PMN formation [142].

Furthermore, EVs and extracellular particles derived from tumor tissue and plasma of OS patients and healthy controls identified six plasma proteins as biomarkers to distinguish cancer-patient plasma from control-subject plasma. These proteins—actin, skeletal alpha (α)-actin (ACTA1), gamma-enteric smooth muscle (ACTG2), a disintegrin and metalloproteinase with thrombospondin motifs 13 (ADAMTS13), hepatocyte growth factor activator (HGFAC), neprilysin (MME), and tenascin C (TNC)—were found only in the plasma isolated from OS patients [143]. Further research is necessary to strengthen our understanding of the role of EVs in OS tumor progression and metastasis. 

### 2.8. Hypoxia and the Tumor Microenvironment of Osteosarcoma

Hypoxia, characterized by reduced oxygen levels within the TME, arises when rapid tumor expansion outpaces the vascular supply’s capacity to deliver oxygen [144]. This scenario activates various cellular mechanisms, primarily orchestrated by the transcriptional regulator Hypoxia-inducible factor-1α (HIF-1α). HIF-1α, under hypoxic conditions, governs cellular responses pivotal to tumorigenesis and metastasis. In a compelling study, Zhang et al. elucidated the profound impact of a hypoxia-immune-related microenvironment on OS prognosis. They delineated how hypoxic conditions actively modulate immune cell infiltration, a phenomenon with significant ramifications for patient outcomes, and identified seven hypoxia- and immune-related genes, including *BNIP3*, *SLC38A5*, *SLC5A3*, *CKMT2*, *S100A3*, *CXCL11*, and *PGM1*, associated with the prognostic signature [145]. Fu et al. crafted a hypoxia-associated prognostic signature directly correlated with OS metastasis and immune infiltration [146]. Their findings offer insights into hypoxia’s multifaceted role, demonstrating its capacity to drive metastasis while simultaneously reshaping the immune landscape. Addressing hypoxia’s therapeutic potential, Chen et al. proposed an innovative strategy. They introduced Polydopamine-coated UiO-66 nanoparticles loaded with therapeutic agents designed explicitly for hypoxia-activated OS therapy [147]. Their approach capitalizes on the hypoxic environment, ensuring targeted drug activation within the tumor region. Furthermore, the intricate interplay between hypoxia and cellular signaling pathways has been spotlighted. For instance, He et al. showcased that zinc oxide nanoparticles could be harnessed to inhibit OS metastasis. They achieved this by perturbing the HIF-1α/BNIP3/LC3B-mediated mitophagy pathway, underscoring the therapeutic potential of targeting hypoxia-driven molecular routes [148]. Integrating therapy modalities, Zhao et al. elegantly demonstrated the enhanced efficacy achieved by coupling a hypoxia inhibitor with conventional chemotherapeutic agents. Their approach bolstered both antitumor and antimetastatic responses against OS, emphasizing the potential of targeting the hypoxic TME to improve therapeutic outcomes [149]. Deepening our understanding of the molecular intricacies of hypoxia, Han et al. conducted an extensive analysis of hypoxia-associated genes in OS [150]. Their results underscored the prognostic significance while highlighting their influence on the immune status and potential therapeutic avenues. Further illuminating hypoxia’s diverse impact, Shen et al. identified a feedback mechanism involving circular RNA Hsa_circ_0000566 and HIF-1α. This interaction fosters OS progression and promotes a shift in cellular metabolism towards glycolysis [151]. Highlighting the protective mechanisms tumors employ, Lu et al. discovered that in hypoxic OS cells, FOXO3a-driven upregulation of HSP90 can shield cells from cisplatin-induced apoptosis. This is achieved by stimulating FUNDC1-mediated mitophagy, emphasizing the challenges posed by hypoxia in cancer therapy [152]. In a recent contribution by Zheng et al., hypoxia’s role in metabolic reprogramming was spotlighted. Their work demonstrated that suppressing lncRNA DLGAP1-AS2 hinders OS progression by inhibiting aerobic glycolysis through the miR-451a/HK2 axis [153]. Recently, Subasinghe mapped hypoxia in real time in in vivo orthotopic syngeneic tumors to improve radiographic mapping, which is critical to the study of cancer and a wide range of diseases [154]. In summation, the hypoxic TME in OS emerges as a central player, orchestrating tumor progression, metastasis, immune modulation, metabolic shifts, and therapeutic responses. Delving into its intricacies offers a promising frontier for innovative therapeutic strategies in OS.

### 2.9. Epithelial-to-Mesenchymal Transformation and the Osteosarcoma Tumor Microenvironment

Epithelial-to-mesenchymal transition (EMT) is a highly complex and regulated pathway where the phenotype of the epithelial cells transforms into mesenchymal characteristics [155]. As OS cells originate from the mesenchymal cells, EMT pathways in OS are more likely associated with maintaining and promoting the mesenchymal phenotype. Gong et al. reported seven novel EMT-related genes for OS outcome prediction, including *CDK3*, *MYC*, *UHRF2*, *STC2*, *COL5A2*, *MMD*, and *EHMT2*. A high expression of EMT-related genes was associated with poor overall and recurrence-free survival compared to the low-expressing group [156]. The GSVA and GSEA analysis showed a nine-gene set for EMT, including *LAMA3*, *LGALS1*, *SGCG*, *VEGFA*, *WNT5A*, *MATN3*, *ANPEP*, *FUCA1*, and *FLNA*, as independent prognostic markers for OS patient survival [157]. Peng et al. established a positive correlation between the expression of EMT-related genes and immunity using GESA and microarray data analysis. Higher EMT gene expression was associated with poor overall and recurrence-free survival [158]. The expression of EMT transcriptional factors, including *TWIST1*, *SLUG*, *ZEB1*, *ZEB2*, and *SNAIL*, is associated with aggressive behavior and metastasis of OS [155,159,160]. Several miRNAs, including miR-18a-5p, miR-22, miR-33A, miR-221, miR-300, miR-580-3p, and miR-610, are associated with the regulation of the EMT in OS cells via targeting TWIST1 [161,162,163,164,165,166]. Chen et al. demonstrated that HDAC5 promotes OS tumor progression by upregulating TWIST 1 expression [167]. The overexpression of ZEB1 is associated with OS tumorigenesis and metastasis, and ZEB1 blocking can suppress metastasis and prevent OBs’ differentiation into OS cells [168,169]. ZEB1/2 suppression by long non-coding (lnc)-RNA sprouty receptor tyrosine kinase signaling antagonist 4-intronic transcript 1 (SPRY4-IT1), mediated through miR-101, showed antitumor activities in OS [170]. Overexpression of SNAIL-1 was found to be associated with the OS progression, mediated through the suppression of E-cadherin expression [171]. *SIRT2* knockdown inhibited SNAIL expression and abrogated the migration-and-invasion-promoting phenotype in OS [172]. Glaucocalyxin A, a bioactive ent-kauranoid diterpenoid, inhibits OS lung metastasis by suppressing the protein expression of SNAIL and SLUG [173]. At the same time, GATA3 suppresses OS progression and metastasis through SLUG regulation [174]. Studies have shown that EMT signaling is vital for maintaining the mesenchymal status of OS. It is also associated with tumor progression, metastasis, and therapeutic responses, so targeting EMT pathways may provide new opportunities to identify potential molecular targets for OS.

### 2.10. Ferroptosis and the Osteosarcoma Tumor Microenvironment

Ferroptosis is a type of nonapoptotic cell death that is dependent on intracellular iron, responsible for the accumulation of lethal lipid reactive oxygen species (ROS) [175]. The mechanism of ferroptosis involves iron-dependent phospholipid peroxidation of polyunsaturated fatty acid chains. The enzyme glutathione peroxidase 4 (GPX4) opposes ferroptosis by inhibiting phospholipid peroxidation [176]. The induction of ferroptosis has been shown to be effective against rhabdomyosarcoma tumor cells [177]. Similarly, therapeutic approaches meant to initiate ferroptosis can be effective against OS tumors [178]. As an example, the transcription activator signal transducer and activator of transcription 3 (STAT3) is known to be activated in many cancers, including OS, and contributes to cisplatin resistance; treatment of OS cisplatin-resistant cell lines in vitro with cisplatin along with a STAT3 inhibitor (BP-1-102) resulted in increased ferroptosis activity and decreased cell viability. Furthermore, treatment with the ferroptosis agonists erastin and RSL3 decreased the cell viability of cisplatin-resistant cell lines following exposure to cisplatin [179]. Thus, activating ferroptosis can be a reasonable approach to minimize cisplatin resistance in OS. Lei et al. developed a prognostic model using ferroptosis-related genes to predict the risk and prognosis of OS patients. In addition, patients in the high-risk group displayed a lower abundance of immune cells, such as dendritic cells, macrophages, B cells, and CD4^+^ T cells [180]. The strongest independent prognostic risk genes related to ferroptosis and immune pathways were found to be WNT16, GLB1L2, CORT, and WAS [181]. Ferroptosis presents a potential therapeutic target against OS and various other types of cancer.

## 3. Tumor Microenvironment Modulating and Targeting Therapies in Osteosarcoma

Historically, the treatment of OS was primarily surgical, with amputation as the standard approach. However, neoadjuvant chemotherapy significantly improved patient outcomes in the 1970s and 1980s, increasing the 5-year survival rate from less than 20% to 60–70% [182,183]. Despite these advances, the survival rate plateaued in the mid-1980s, and further improvements have been elusive [184]. Improvements in patient outcomes, especially for high-risk patients with metastases, have been limited. This stagnation is primarily due to the complex genetic heterogeneity of OS, which makes developing effective targeted therapies difficult. In addition, OS is an aggressive disease often associated with metastases at diagnosis, further complicating the therapeutic landscape [182,185]. This led to a shift in focus toward understanding the biology of OS and identifying novel therapeutic targets. 

In recent years, immunotherapy has emerged as a promising approach to fight against OS. Various immunotherapies are currently being explored in clinical trials, including immune checkpoint inhibitors, cancer vaccines, and adoptive cell therapies (Table 1). Several examples of immunotherapy clinical trials are presented below. 

Immune checkpoint inhibitors: In OS, the PD-1/PD-L1 pathway is often upregulated, leading to immune evasion. Blocking this pathway with anti-PD-1 or anti-PD-L1 antibodies can enhance the immune response against OS cells [186]. A study of pembrolizumab in patients with relapsed or metastatic OS not eligible for curative surgery has been conducted (NCT03013127). Another trial is investigating the use of avelumab in patients with recurrent or progressive OS (NCT03006848). 

Cancer vaccines aim to brace the immune system to attack specific cancer-associated antigens. In OS, several tumor-associated antigens have been identified and are being targeted in vaccine trials. A study recruited participants to evaluate RNA-lipid particle (RNA-LP) vaccines for recurrent pulmonary OS (NCT05660408). Adoptive cell therapies use genetically engineered immune cells to target cancer cells. CAR-T cell therapy, which uses T cells engineered to express a chimeric antigen receptor (CAR) that recognizes and targets a specific antigen in cancer cells, is one of the most promising forms of adoptive cell therapy. A study evaluated the safety and efficacy of fourth-generation engineered CAR T cells targeting sarcomas, including OS (NCT03356782). Despite the promising aspect of these approaches, immunotherapy’s efficacy in OS has been variable, and many patients do not respond to treatment. This has led to the exploration of combination therapies, which use multiple treatments to target different aspects of the cancer TME. For example, a study recruited participants to test the combination of atezolizumab and cabozantinib for treating adolescents and young adults with recurrent or metastatic OS (NCT05019703).

In conclusion, while significant challenges remain in treating OS, immunotherapy has opened new avenues for improving patient outcomes. Clinical trials and translational research are crucial to refine these strategies further and bring us closer to a cure for high-risk patients. T cell-based immunotherapies, including chimeric antigen receptor (CAR) T cell therapy, have shown promise in preclinical models of OS [187,188]. For example, Wang et al. developed an anti-CD166/4-1BB CAR-T cell therapy that showed potent antitumor activity against OS [189]. Similarly, Lin et al. discussed the potential of CAR-T cell therapy targeting GD2 or HER2 in OS [188]. Wang et al. showed that alternative human γδ T cells induce CD8^+^ T cell antitumor responses via antigen presentation through the HSP90-MyD88-JNK pathway [190]. B7-H3 targeted CAR-T cells had high antitumor efficacy against OS in vitro and in vivo [191]. In another study, the human erythropoietin-producing hepatocellular receptor tyrosine kinase class A2 (EphA2)-directed CAR-T cells showed antitumor effects against human OS and Ewing sarcoma tumor cells both in vitro and in vivo. Systemic infusion of EphA2 CAR-T cells traffic to and eradicate tumor deposits in murine liver and lung metastases [192].

## 4. Conclusions and Future Directions

OS patient survival dramatically decreases with therapeutic resistance and metastatic disease, occurring most frequently in the lungs. OS is a heterogeneous and complex disease that is still largely understudied. The TME of OS is a complex network of cellular and acellular components designed by tumor cells to promote successful progression and metastasis. OS is a highly heterogenous and metastatic bone tumor that remains challenging to treat, particularly for metastatic patients. A deeper understanding of the complex dynamics of the OS TME, particularly the diversity of immune cells, real-time cell–cell communication, and TME components, will help to design better therapeutic options that are currently lacking for OS patients. Recently, spatial multiomics technologies have been used to understand the tumor immune cell communication in the TME of OS. Mathematical models are also being implemented to understand the immune and cancer cell interactions and the effect of colocalization on OS tumor growth [193,194]. The challenges associated with OS therapy are limited T cell infiltration and an abundance of immunosuppressive cells, particularly M2 macrophages, MDSCs, and Treg cells. There is a high potential for reprogramming the TME components of OS to overcome the prevalent issue of treatment and immunotherapeutic resistance. 

In this review, various aspects of the OS TME, including cellular and acellular components, their communication, and their therapeutic possibilities, have been described. Finding novel immune and nonimmune cell markers, discovering more specific therapeutic targets, determining interactions among cellular and noncellular components of the OS TME, and using multidisciplinary knowledge and combination therapy will improve OS patient outcomes.

## Figures and Tables

**Figure 1 cancers-15-05108-f001:**
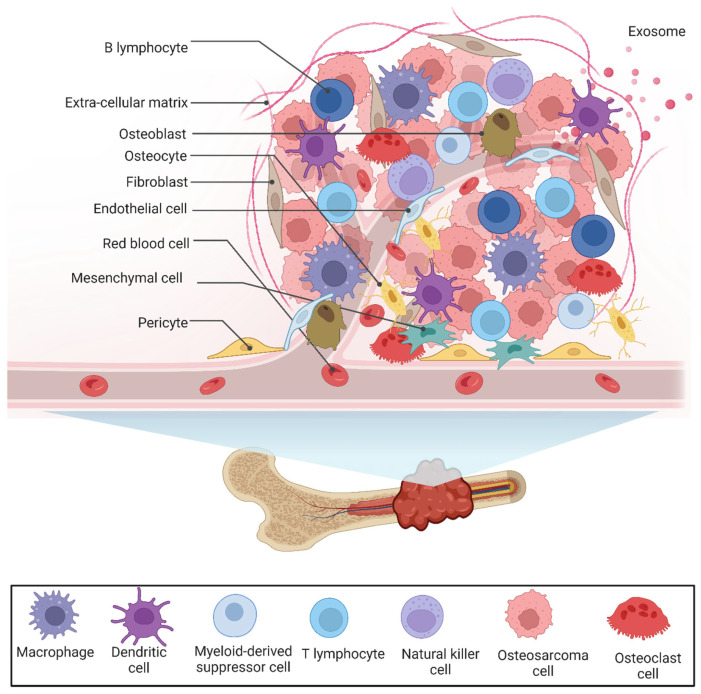
Cellular and noncellular components of OS tumor microenvironment.

**Figure 2 cancers-15-05108-f002:**
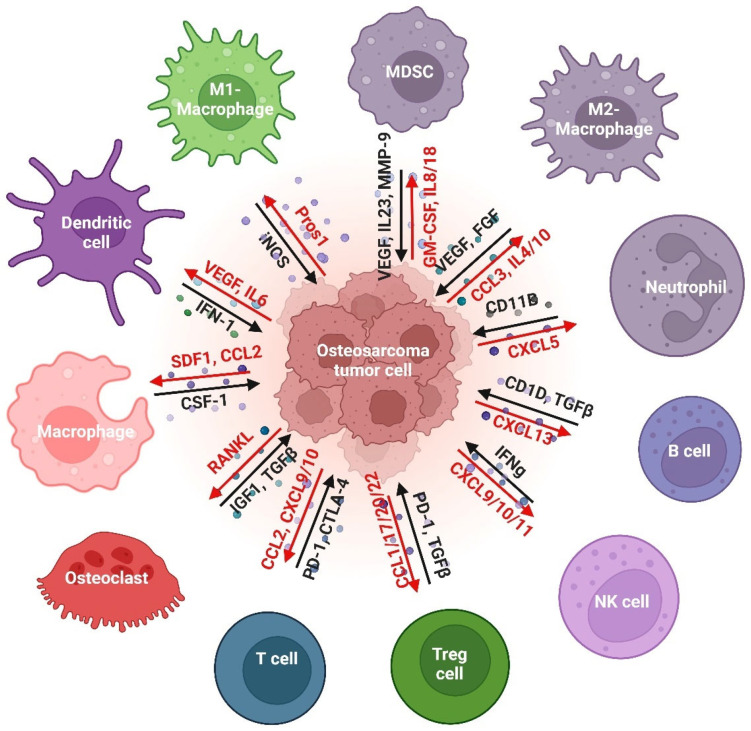
Tumor and immune cell interactions in OS tumor microenvironment.

**Table 1 cancers-15-05108-t001:** Various therapies targeting tumor microenvironment of osteosarcoma.

	Name of Compound	Target Pathway	Mechanism of Action	Phase	NCT Number
1.	Cytokine-induced killer (CIK)	Adoptive immunotherapy	Cytokine-induced killer	I	NCT03782363
2.	Multicomponent immune-based therapy (MKC1106-PP)	Immunotherapy	T-cell	I	NCT00423254
3.	Haploidentical transplant and donor natural killer cells	Immunotherapy	NK cell activation	II	NCT02100891
4.	GD2-targeted modified T-cells (GD2 CAR-T)	Immunotherapy	CAR-T	I	NCT02107963
5.	Activated T cells armed with GD2 bispecific antibody	Immunotherapy	CAR-T	I/II	NCT02173093
6.	Pembrolizumab	Immunotherapy	PD1	II	NCT02301039
7.	Nivolumab plus ipilimumab	Immunotherapy	PD1 and CTLA-4	I/II	NCT02304458
8.	Nivolumab plus ipilimumab	Immunotherapy	PD1 and CTLA-4	II	NCT02500797
9.	Durvalumab plus tremelimumab	Immunotherapy	PD1 and CTLA-4	II	NCT02815995
10.	Nivolumab plus ipilimumab	Immunotherapy	PD1 and CTLA-4	II	NCT02982486
11.	Avelumab	Immunotherapy	PDL1	II	NCT03006848
12.	Avelumab	Immunotherapy	PDL1	II	NCT03006848
13.	Pembrolizumab	Immunotherapy	PD1	II	NCT03013127
14.	Haploidentical donor NK cells and Hu14.18-IL2	Immunotherapy	NK cell activation	I	NCT03209869
15.	NKTR-214 and nivolumab	Immunotherapy	PD1 and IL2 agonist	II	NCT03282344
16.	Sarcoma-specific CAR-T cells	Immunotherapy	CAR-T	I/II	NCT03356782
17.	CAB-AXL-ADC plus PD-1 inhibitor	Immunotherapy	PD1	I/II	NCT03425279
18.	EGFR806 CAR-T cells	Immunotherapy	EGFR	I	NCT03618381
19.	C7R-GD2 CAR-T cells	Immunotherapy	CAR-T	I	NCT03635632
20.	Vigil	Engineered autologous tumor cell immunotherapy	GM-CSF, TGFβ-1, and TGFβ-2		NCT03842865
21.	Famitinib plus camrelizumab	RTK and checkpoint inhibition	VEGFR-2, -3 and FGFR-1, -2, -3, -4 and PD1/PDL1	II	NCT04044378
22.	Camrelizumab (PD1) with neoadjuvant chemotherapy	Checkpoint inhibition	PD1	II	NCT04294511
23.	CAR-T plus chemotherapy	Immunotherapy	CAR-T cells and sarcoma vaccines	I/II	NCT04433221
24.	B7H3 CAR-T cells	Immunotherapy	CAR-T, B7H3-specific receptor	I	NCT04483778
25.	GD2-targeted modified T-cells (GD2 CAR-T)	Immunotherapy	CAR-T	I	NCT04539366
26.	Oleclumab plus durvalumab	Immunotherapy	Anti-CD73 monoclonal antibody and PD-1	II	NCT04668300
27.	B7H3 CAR-T cells	Immunotherapy	CSRT, B7H3-specific receptor	I	NCT04897321
28.	Atezolizumab and cabozantinib	Immunotherapy	Immune checkpoint inhibitor	II	NCT05019703
29.	Tislelizumab	Immunotherapy	PD1	II	NCT05241132
30.	Nivolumab plus ipilimumab	Immunotherapy	PD1 and CTLA-4	II	NCT05302921
31.	RNA-lipid particle vaccines	Cancer vaccine	Cancer vaccine	I/II	NCT05660408

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
