# Peer review of "Decoding the Impact of Tumor Microenvironment in Osteosarcoma Progression and Metastasis"

_cancers, 2023, doi:10.3390/cancers15205108_

Round 1
Reviewer 1 Report
In this paper, authors investigate the role of different immune cell types in the progression of osteosarcoma. The paper is well written, and it provides a great insight about osteosarcoma’s tumor microenvironemt.
The paper discusses numerous studies that primarily center around the influence of individual cell types on tumor progression. Nevertheless, it is imperative to underscore that computational and mathematical models elucidate how the dynamic interplay among various cell types significantly impacts tumor progression, rather than any single cell type or marker in isolation. Here are some examples:
· Le, Trang, Sumeyye Su, Arkadz Kirshtein, and Leili Shahriyari. "Data-driven mathematical model of osteosarcoma." Cancers 13, no. 10 (2021): 2367.
· Hu, Yu, Navid Mohammad Mirzaei, and Leili Shahriyari. "Bio-Mechanical Model of Osteosarcoma Tumor Microenvironment: A Porous Media Approach." Cancers 14, no. 24 (2022): 6143.
To enhance the paper, the authors should place more emphasis on this pivotal insight.
Author Response
We appreciate the reviewer's suggestion, and we believe that understanding cell-cell communication and the spatial distribution of cells will improve our knowledge. However, there is still a paucity of literature available on it in osteosarcoma. We incorporated literature, including the mathematical and computational model addressing the dynamic interplay among various cell types, as suggested by the reviewer in the conclusion and future direction section.
“Recently, spatial multi-omics technologies have been used to understand the tumor immune cell communication in the TME of OS. Mathematical models are also being implemented to understand the immune and cancer cell interactions and the effect of colocalization on OS tumor growth [192,193]”.
Reviewer 2 Report
Well done review by Nirala et al. addressing cellular constituents of osteosarcoma tumor microenvironment. Overall, limited comments or concerns, except as follows:
1) Impact of hypoxia in bone marrow niche not discussed. This is a key variable in tumor microenvironment in most solid tumors, impacting metabolic milieu and immune function, but also critical to hematopoietic niche of bone marrow microenvironment. Although not exhaustive, there are studies addressing this.
2) Figure 2 is too simplistic as it under-represents the multiplicity of chemokines, cytokines, and receptor-ligand combinations involved in these interactions as well as the direct immune cell interactions and plasticity at work.
3) More information should be provided on how these cell types are defined and how variability in defining cellular constituents across experimental readouts (flow cytometry, IHC, sequencing) can impact data obtained and conclusions drawn. This is especially true for TANs, macrophages, and mast cells.
Author Response
- Impact of hypoxia in bone marrow niche not discussed. This is a key variable in tumor microenvironment in most solid tumors, impacting metabolic milieu and immune function, but also critical to hematopoietic niche of bone marrow microenvironment. Although not exhaustive, there are studies addressing this.
We appreciate the reviewer's suggestion. We also believe it is crucial to understand the role of hypoxia in the TME of OS. As suggested, we incorporated a separate sub-heading, “ 2.8. Hypoxia and tumor microenvironment of osteosarcoma’ in the manuscript.
Hypoxia, characterized by reduced oxygen levels within the TME, arises when rapid tumor expansion outpaces the vascular supply's capacity to deliver oxygen [149]. This scenario activates various cellular mechanisms, primarily orchestrated by the transcriptional regulator hypoxia-inducible factor-1α (HIF-1α). HIF-1α, under hypoxic conditions, governs cellular responses pivotal to tumorigenesis and metastasis. In a compelling study, Zhang et al., elucidated the profound impact of a hypoxia-immune-related microenvironment on OS's prognosis. They delineated how hypoxic conditions actively modulate immune cell infiltration, a phenomenon with significant ramifications for patient outcomes, and identified seven hypoxia- and immune-related genes, including BNIP3, SLC38A5, SLC5A3, CKMT2, S100A3, CXCL11 and PGM1 associated with the prognostic signature [150]. Fu et al., crafted a hypoxia-associated prognostic signature directly correlating with OS metastasis and immune infiltration [151]. Their findings offer insights into hypoxia's multifaceted role, demonstrating its capacity to drive metastasis while simultaneously reshaping the immune landscape. Addressing hypoxia's therapeutic potential, Chen et al., proposed an innovative strategy. They introduced Polydopamine-coated UiO-66 nanoparticles loaded with therapeutic agents designed explicitly for hypoxia-activated OS therapy [152]. Their approach capitalizes on the hypoxic environment, ensuring targeted drug activation within the tumor region. Furthermore, the the interplay between hypoxia and cellular signaling pathways has been spotlighted. For instance, He et al., showcased that zinc oxide nanoparticles could be harnessed to inhibit OS metastasis. They achieve this by perturbing the HIF-1α/BNIP3/LC3B-mediated mitophagy pathway, underscoring the therapeutic potential of targeting hypoxia-driven molecular routes [153]. Integrating therapy modalities, Zhao et al., elegantly demonstrated the enhanced efficacy achieved by coupling a hypoxia inhibitor with conventional chemotherapeutic agents. Their approach bolstered both antitumor and antimetastatic responses against OS, emphasizing the potential of targeting the hypoxic TME to improve therapeutic outcomes [154]. Deepening our understanding of the molecular intricacies of hypoxia, Han et al., conducted an extensive analysis of hypoxia-associated genes in OS [155]. Their results underscored the prognostic significance while highlighting their influence on the immune status and potential therapeutic avenues. Further illuminating hypoxia's diverse impact, Shen et al., identified a feedback mechanism involving circular RNA Hsa_circ_0000566 and HIF-1α. This interaction fosters OS progression and promotes a shift in cellular metabolism towards glycolysis [156]. Highlighting the protective mechanisms tumors employ, Lu et al., discovered that in hypoxic OS cells, FOXO3a-driven up-regulation of HSP90 can shield cells from cisplatin-induced apoptosis. This is achieved by stimulating FUNDC1-mediated mitophagy, emphasizing the challenges posed by hypoxia in cancer therapy [157]. In a recent contribution by Zheng et al., hypoxia's role in metabolic reprogramming was spotlighted. Their work demonstrated that suppressing lncRNA DLGAP1-AS2 hinder OS progression by inhibiting aerobic glycolysis through the miR-451a/HK2 axis [158]. Recently, Subasinghe, mapped hypoxia in realtime in in vivo orthotopic syngeneic tumor to improve the ability to radiographic mapping, which is critical to the study of cancer and a wide range of diseases [159]. In summation, the hypoxic TME in OS emerges as a central player, orchestrating tumor progression, metastasis, immune modulation, metabolic shifts, and therapeutic responses. Delving into its intricacies offers a promising frontier for innovative therapeutic strategies in OS.
2) Figure 2 is too simplistic as it under-represents the multiplicity of chemokines, cytokines, and receptor-ligand combinations involved in these interactions as well as the direct immune cell interactions and plasticity at work.
We appreciate the reviewer for raising this concern and we have modified Figure 2 accordingly.
3) More information should be provided on how these cell types are defined and how variability in defining cellular constituents across experimental readouts (flow cytometry, IHC, sequencing) can impact data obtained and conclusions drawn. This is especially true for TANs, macrophages, and mast cells.
We appreciate the reviewer for this suggestion. We provided the cell type information in the manuscript where needed and mentioned in the literature.
Reviewer 3 Report
This is an interesting and well-written review article; however, I have 2 suggestions as following:
1. Please discuss about the regulation of the epithelial to mesenchymal transition in osteosarcoma microenvironment.
2. Please discuss about the interaction between ferroptosis and immunity in the osteosarcoma microenvironment.
Minor editing of English language required.
Author Response
- Please discuss about the regulation of the epithelial to mesenchymal transition in the osteosarcoma microenvironment.
We appreciate the reviewer's concern and incorporated a separate subheading on EMT and OS to address it.
2.9 Epithelial to mesenchymal transformation and osteosarcoma tumor
“Epithelial-to-mesenchymal transition (EMT) is a highly complex and regulated pathway where the phenotype of the epithelial cells transforms into mesenchymal characteristics [160]. As OS cells originate from the mesenchymal cells, EMT pathways in OS are more likely associated with maintaining and promoting the mesenchymal phenotype. Gong et al., reported seven novel EMT-related genes for OS outcome prediction, including CDK3, MYC, UHRF2, STC2, COL5A2, MMD, and EHMT2. A high expression of EMT-related genes was associated with poor overall and recurrence-free survival compared to the low-expressing group [161]. The GSVA and GSEA analysis showed a 9-genes set of EMT, including LAMA3, LGALS1, SGCG, VEGFA, WNT5A, MATN3, ANPEP, FUCA1, and FLNA as an independent prognostic marker for OS patients survival [162]. Peng et al., established a positive correlation between the expression of EMT-related genes and immunity using GESA and microarray data analysis. Higher EMT gene expression was associated with poor overall and recurrence-free survival [163]. The expression of EMT transcriptional factors, including TWIST1, SLUG, ZEB1, ZEB2, and SNAIL, are associated with aggressive behavior and metastasis of OS [164-166]. Several miRNAs, including miR-18a-5p, miR-22, miR-33A, miR-221, miR-300, miR-580-3p, and miR-610, are associated with the regulation of the EMT in OS cells via targeting TWIST1 [167-172]. Chen et al., demonstrated that HDAC5 promotes OS tumor progression by upregulating TWIST 1 expression [173]. The overexpression of ZEB1 is associated with OS tumorigenesis and metastasis, and ZEB1 blocking can suppress metastasis and prevent OBs differentiation into OS cells [174,175]. ZEB1/2 suppression by long non-coding (lnc)-RNA sprouty receptor tyrosine kinase signaling antagonist 4-intronic transcript 1 (SPRY4-IT1), mediated through miR-101 shown antitumor activities in OS [176]. Overexpression of SNAIL‑1 was found to be associated with the OS progression, mediated through the suppression of E‑cadherin expression [177]. SIRT2 knockdown inhibited SNAIL degradation and abrogated their migration and invasion-promoting phenotype in OS [178]. Glaucocalyxin A, a bioactive ent-kauranoid diterpenoid, inhibits OS lung metastasis by suppressing the protein expression of SNAIL and SLUG [179]. At the same time, GATA3 suppresses OS's progression and metastasis through SLUG regulation [180]. Studies have shown that EMT signaling is vital for maintaining the mesenchymal status of OS. It is also associated with tumor progression, metastasis, and therapeutic responses, so targeting EMT pathways may provide new opportunities to identify potential molecular targets for OS.
- Please discuss about the interaction between ferroptosis and immunity in the osteosarcoma microenvironment.
We appreciate the reviewer suggestion and incorporate a separate subheading entitled Ferroptosis and the OS TME.
Ferroptosis is a type of non-apoptotic cell death that is dependent on intracellular iron, responsible for the accumulation of lethal lipid reactive oxygen species (ROS) [181]. The mechanism of ferroptosis involves iron-dependent phospholipid peroxidation, ROS and phospholipids with polyunsaturated fatty acid chains. The enzyme glutathione peroxidase 4 (GPX4) opposes ferroptosis by inhibiting phospholipid peroxidation [182]. The induction of ferroptosis has been shown to be effective against rhabdomyosarcoma tumor cells [183]. Similarly, therapeutic approaches meant to initiate ferroptosis can be effective against OS tumors [184]. As an example, the transcription activator signal transducer and activator of transcription 3 (STAT3) is known to be activated in many cancers, including OS, and contributes to cisplatin resistance; treatment of OS cisplatin-resistant cell lines in vitro with cisplatin along with a STAT3 inhibitor (BP-1-102) resulted in increased ferroptosis activity and decreased cell viability. Furthermore, treatment with ferroptosis agonists Erastin and RSL3 decreased cell viability of cisplatin-resistant osteosarcoma cell lines following exposure to cisplatin [185]. As such, activating ferroptosis can be a reasonable approach to minimize cisplatin resistance in OS. Lei et al., developed a prognostic model using ferroptosis-related genes to predict the risk and prognosis of OS patients. In addition, patients in the high-risk group displayed a lower abundance of immune cells, such as dendritic cells, macrophages, B cells, and CD4+ T cells [186]. The strongest independent prognostic risk genes, related to ferroptosis and immune pathways, were found to be WNT16, GLB1L2, CORT, and WAS [187]. Ferroptosis presents a potential therapeutic target against OS and various other types of cancer.